# SON protects nascent transcripts from unproductive degradation by counteracting DIP1

**Mandy Li-Ian Tay**[1], **Jun Wei Pek**[1,2]*

**1** Temasek Life Sciences Laboratory, Singapore, Singapore, **2** Department of Biological Sciences, National University of Singapore, Singapore

* junwei@tll.org.sg

**Data Availability Statement:** All relevant data are within the manuscript and its Supporting Information files.

**Funding:** JWP and MT are funded by the Temasek Life Sciences Laboratory. The funders had no role

## Abstract

Gene expression involves the transcription and splicing of nascent transcripts through the removal of introns. In *Drosophila*, a double-stranded RNA binding protein Disco-interacting protein 1 (DIP1) targets INE-1 stable intronic sequence RNAs (sisRNAs) for degradation after splicing. How nascent transcripts that also contain INE-1 sequences escape degradation remains unknown. Here we observe that these nascent transcripts can also be bound by DIP1 but the *Drosophila* homolog of SON (Dsn) protects them from unproductive degradation in ovaries. Dsn localizes to the satellite body where active decay of INE-1 sisRNAs by DIP1 occurs. Dsn is a repressor of DIP1 posttranslational modifications (primarily sumoylation) that are assumed to be required for efficient DIP1 activity. Moreover, the pre-mRNA destabilization caused by Dsn depletion is rescued in DIP1 or Sumo heterozygous mutants, suggesting that Dsn is a negative regulator of DIP1. Our results reveal that under normal circumstances nascent transcripts are susceptible to DIP1-mediated degradation, however intronic sequences are protected by Dsn until intron excision has taken place.

## Author summary

During transcription, nascent RNAs are exposed to various RNA degradation machineries in the nucleus. Nascent RNAs undergo a process called splicing that removes noncoding sequences (known as introns) in order to produce protein-coding messenger RNAs. In the vinegar fly *Drosophila*, introns that contain a transposable sequence known as INE-1 are recognized and degraded by a protein called DIP1. This process usually happens after splicing so that DIP1 does not degrade nascent RNAs. How such a target specificity and temporal control are achieved is not known. Here we found that nascent RNAs are already being recognized by DIP1. However, its activity is inhibited by the SON protein that also binds to nascent RNAs. After splicing, the inhibition of DIP1 by SON is relieved, allowing a spatial and temporal control of DIP1 activity. This regulation is important as it prevents unspecific decay of nascent RNAs that can drastically affect gene expression.

in study design, data collection and analysis, decision to publish, or preparation of the manuscript.

**Competing interests:** The authors have declared that no competing interests exist.

# Introduction

The first few steps of gene expression include the production of nascent transcripts and the removal of introns via the splicing reaction. The nucleus contains numerous RNA decay machineries, and thus nascent transcripts need to be protected by various mechanisms to ensure productive gene expression [1]. Certain intronic sequences in the excised introns can target them for degradation. In principle, nascent transcripts that also contain the same intronic sequences can also be subjected to decay. In budding yeast, certain double-stranded RNA (dsRNA) stem-loop structures trigger RNase III-mediated degradation of both the excised introns and unspliced pre-mRNAs [2]. Whereas in fission yeast, decay-promoting introns target unspliced pre-mRNAs for degradation by recruiting the exosome specificity factor Mmi1 [3]. For productive gene expression, decay-promoting introns should only trigger degradation of excised introns and not nascent transcripts. How nascent transcripts avoid such degradation is unknown.

Stable intronic sequence RNAs (sisRNAs) are intron-containing transcripts that are relatively more stable than their excised counterparts or those that undergo nonsense-mediated decay [4–9] They have been shown to regulate various biological processes such as germline stem cell (GSC) maintenance and embryonic development [10,11]. The *Drosophila* chromosome four contains an extremely high abundance of INE-1 sequences in the introns [12]. INE-1 belongs to class of transposable element abundant in *Drosophila* [12–14]. As a result, the fourth chromosome is a region where a high density of INE-1 sisRNAs is being produced. Here, a double-stranded RNA binding protein Disco-interacting protein 1 (DIP1) binds and degrades INE-1 sisRNAs [15]. This leads to the formation of microscopically visible DIP1-positive nuclear bodies known as satellite bodies around the fourth chromosomes [15]. DIP1 only degrades INE-1 sisRNAs after splicing as pre-mRNAs containing INE-1 sequences were unaffected in DIP1 mutants [15]. It is not understood how such a target specificity is achieved (Fig 1A).

In this study, we report the conserved protein SON (or Dsn in *Drosophila*) acts to protect nascent transcripts containing INE-1 from being degraded by DIP1. Our results show that nascent transcripts are bound to DIP1. However, the presence of Dsn inhibits DIP1 at the level of RNA decay activity until intron excision is completed. Thus, Dsn acts as a 'timer' to ensure that intronic sequences are only subjected to degradation after splicing in order for productive gene expression to take place.

# Results

## Dsn is a novel satellite body component

We previously reported that Dsn regulates GSCs by repressing the expression of *regena* (*rga*), which encodes for NOT2 (a component of the CCR4-NOT complex) required for the maintenance of GSCs [16–19]. To examine the localization of endogenous Dsn, we attempted to generate antibodies against Dsn using bacteria-expressed GST-tagged Dsn and peptide sequence of Dsn, but were unsuccessful. We therefore generated transgenic flies over-expressing FLAG-Dsn. When driven by a germline driver (*MTD-Gal4*), FLAG-Dsn rescued the *dsn* mutant phenotype (S1 Fig, discussed later), verifying that our FLAG-Dsn transgene produced a fully functional protein. We observed that FLAG-Dsn localized around the presumed fourth chromosomes in the ovarian nurse cells, reminiscent of the satellite body. Co-staining with the satellite body marker DIP1 confirmed that FLAG-Dsn is a satellite body component as both proteins co-localized around the presumed fourth chromosomes in the nurse cell nucleus (Fig 1B, arrowheads). Specificity of the staining was verified by the lack of signals in the somatic

**A**

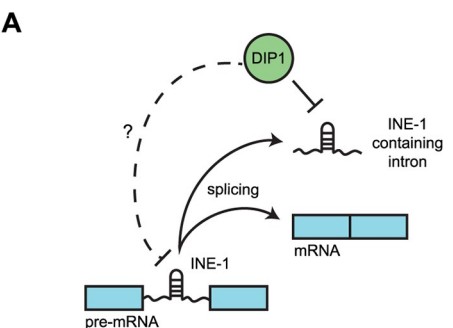

**B**

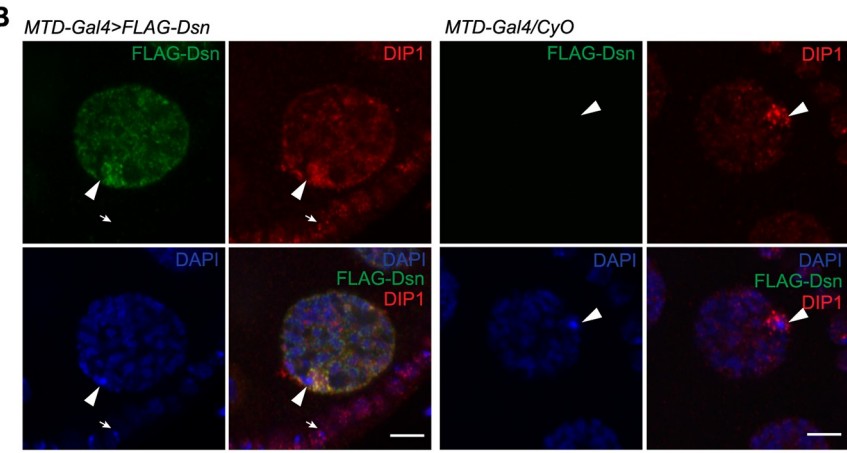

**C** *MTD-Gal4>FLAG-Dsn*

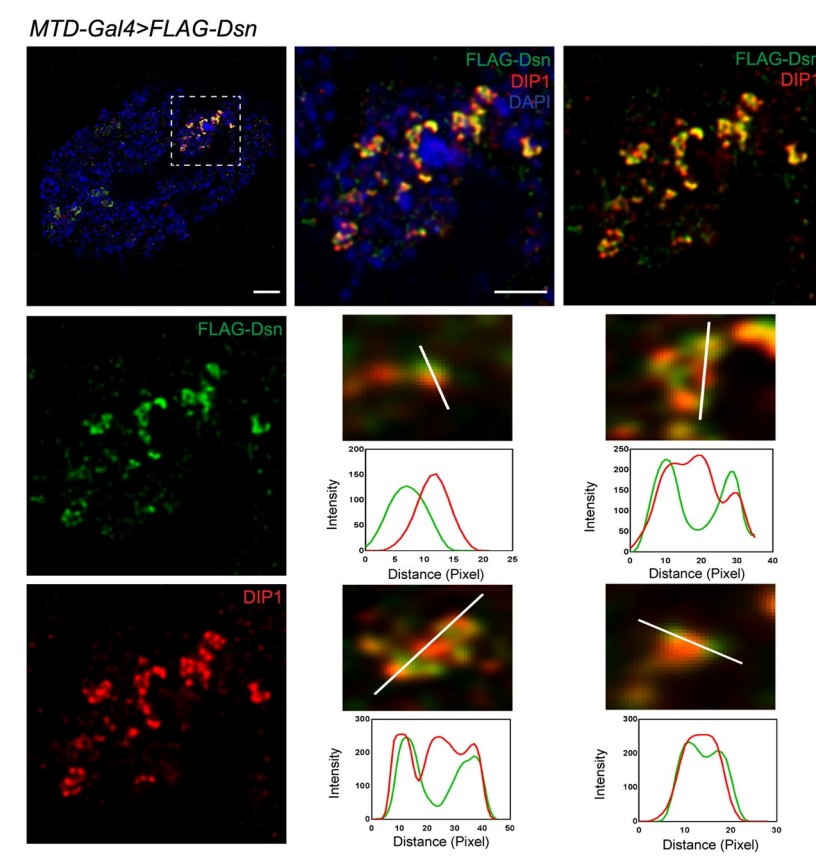

**Fig 1. *Drosophila* Dsn is a satellite body component.** (A) Working model of DIP1 in regulating the expression of *INE-1* containing pre-mRNA and *INE-1* sisRNAs in *Drosophila*. (B) Confocal images showing the localization of FLAG-Dsn (green), DIP1 (red) and DAPI (blue) in a *MTD-Gal4>FLAG-Dsn* nurse cell nucleus. *MTD-Gal4/CyO* served as a negative control. Arrowheads point to the heterochromatin of the fourth chromosomes in the nurse cell nuclei. Arrows point to the heterochromatin of the fourth chromosomes in the follicle cells. Scale bar: 5 μm. (C) Super-resolution confocal microscopy images of a *MTD-Gal4>FLAG-Dsn* nurse cell nucleus stained for FLAG-Dsn (green), DIP1 (red) and DAPI (blue). Inset: magnification of area (dotted box) around the 4th chromosome. Intensity plots showing the intensities of FLAG-Dsn and DIP1 signals at different locations. Scale bar: 20 μm.

follicle cells (Fig 1B, arrows), and the non-transgenic control (*MTD-Gal4/CyO*) (Fig 1B). Although it should be noted that due to over-expression, the localization of FLAG-Dsn may not precisely reflect that of endogenous Dsn, we believe that the protein reflects it endogenous localization due to its ability to rescue the *dsn* mutant phenotype.

To investigate further, we examined the localizations of DIP1 and Dsn more closely by super-resolution deconvolution (STED) microscopy. Under the super-resolution microscope, the localization patterns of DIP1 and FLAG-Dsn were better resolved. Interestingly, DIP1 and FLAG-Dsn did not overlap completely. Fig 1C shows a representative single optical section of the satellite bodies. Four different regions of the satellite bodies are presented. Measurements of signal intensities showed that DIP1 and FLAG-Dsn only partially overlapped, where they appeared associated closely with each other in a network (Fig 1C).

## Dsn promotes the stability of INE-1 containing pre-mRNAs

DIP1 acts to repress INE-1 sisRNAs after splicing as DIP1 mutant ovaries exhibited an increase in INE-1 sisRNAs but no change in the INE-1 containing pre-mRNAs [15]. Since Dsn is also a satellite body component, we examined if Dsn performs the same function as DIP1. We used a mutant allele *CG8273^GS7314^* that contains a transposon insertion at the 5' UTR of *dsn* (or CG8273) leading to a dramatic loss (over 90% reduction) of *dsn* mRNA expression [16]. In contrast to DIP1 mutants, *dsn* mutant ovaries had a decrease in INE-1 sisRNAs and mRNAs from genes harboring INE-1 sequences in their introns (*CG32000* and *CG2316*) (Fig 2A and 2B). To examine whether if pre-mRNA levels were affected, we performed RT-qPCR using primers that flank the exon-intron junctions, and found a consistent decrease in *CG32000*, *CG2316* and *ANK* pre-mRNAs (Fig 2C). In contrast, the pre-mRNAs of genes that do not contain INE-1 sequences (*Maverick* and *Myoglianin*) were unaffected in *dsn* mutant ovaries (Fig 2C). Although there was a decrease in *ANK* pre-mRNA, we did not observe a similar decrease in *ANK* mRNA in *dsn* mutant ovaries, which may suggest a feedback mechanism regulating *ANK* mRNA. Together, these results indicate that Dsn specifically regulates pre-mRNAs containing INE-1.

In *dsn* mutant ovaries, both the levels of pre-mRNAs and mRNAs were down-regulated, therefore excluding the possibility that Dsn regulates splicing. Next, we wondered if Dsn regulates the stability of pre-mRNA. We performed transcription inhibition assays and measured the levels of a relatively abundant pre-mRNA from *CG32000* over time by RT-qPCR. Ovaries were first treated with alpha-amanitin to inhibit transcription for a period of 0.5 hr. We observed that in *dsn* mutant ovaries, the *CG32000* pre-mRNA had a higher magnitude of decrease than in controls (Fig 2D). As a control, we examined the *Marverick* pre-mRNA and did not observe any difference between wild-type and *dsn* mutants (S2 Fig). We repeated the experiment with another transcription inhibitor Actinomycin D under a longer incubation period, and similar results were obtained (Fig 2E), suggesting that pre-mRNA was less stable in *dsn* mutant ovaries. Together, our results suggest that Dsn is required to ensure robust expression of INE-1 containing pre-mRNA, at least in part, by ensuring their stability (Fig 2F).

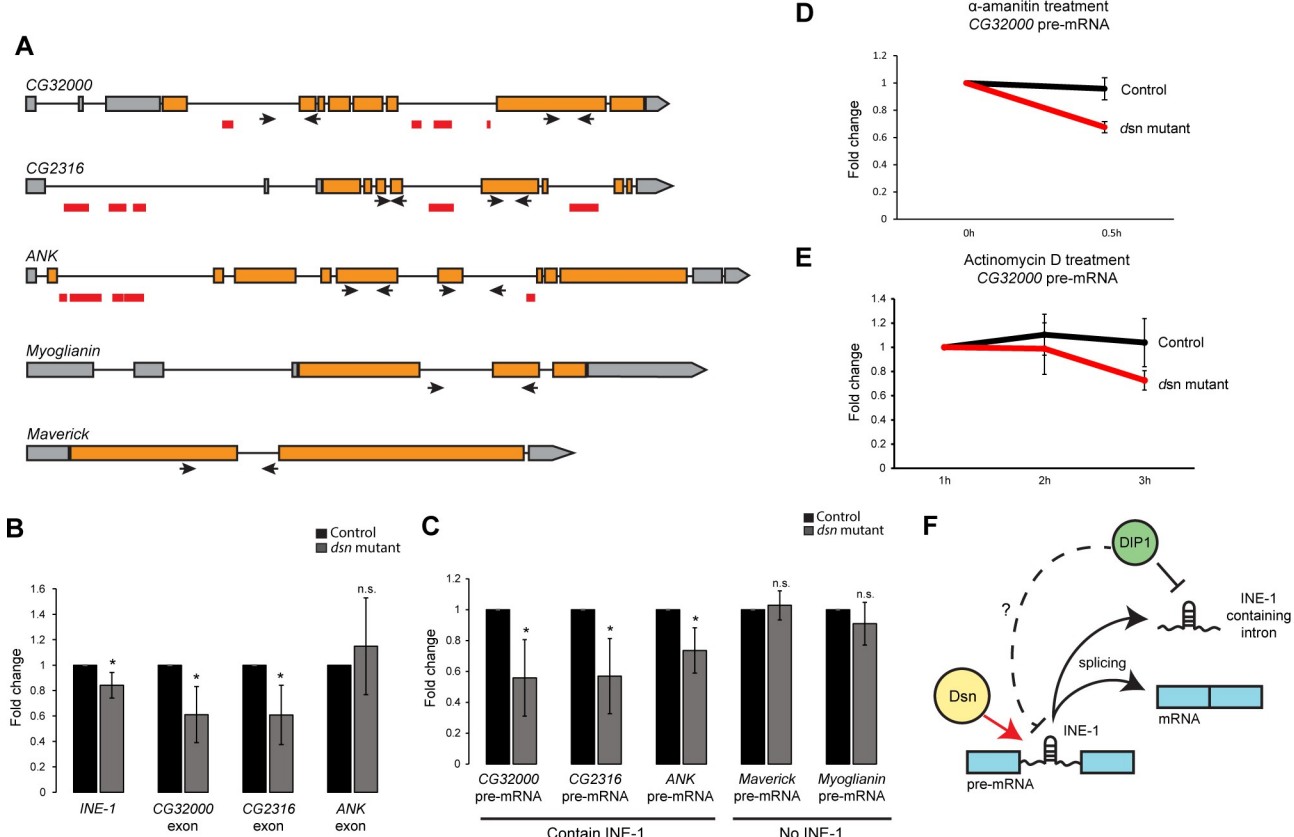

**Fig 2. *Dsn* is required for robust expression of 4ᵗʰ chromosome genes containing intronic *INE-1* elements.** (A) Gene models of 4ᵗʰ chromosome genes, *CG32000*, *CG2316*, *ANK*, *Myoglianin* and *Maverick*. Red bars indicate the locations of *INE-1* elements. Arrows depict the primers used for qPCR. (B) RT-qPCR showing the downregulation of *INE-1*, *CG32000* exon and *CG2316* exon in *dsn* mutant versus control ovaries. N = 3 biological replicates. *p < 0.05, two-tailed t test. N.S., not significant p>0.05. (C) RT-qPCR showing the downregulation of 4ᵗʰ chromosome genes containing *INE-1* in *dsn* mutant but not in control ovaries. N = 3 biological replicates. *p < 0.05, two-tailed t test. N.S., not significant p>0.05. (D) Chart showing the levels of *CG32000* pre-mRNA in the ovaries of control and *dsn* mutant flies before and after 0.5h of α-amanitin treatment. Error bars depict SD from three biological replicates. (E) Chart showing the levels of *CG32000* pre-mRNA in the ovaries of control and *dsn* mutant flies before and after 1h, 2h and 3h of actinomycin D treatment. Error bars depict SD from three biological replicates. (F) Working model of Dsn involved in the regulation of *INE-1* containing pre-mRNA and *INE-1* sisRNA by DIP1.

## DIP1 binds to nascent transcripts in a Dsn-independent manner

It was surprising that although Dsn and DIP1 localize to satellite bodies, *dsn* and DIP1 mutants displayed contrasting phenotypes suggesting different functions. Since Dsn promotes the expression of INE-1 containing pre-mRNAs, and DIP1 had been found to degrade INE-1 sisR-NAs, we explored the possibility that Dsn may inhibit DIP1 from degrading pre-mRNAs.

We checked if Dsn regulates the expression of DIP1. RT-qPCR experiment revealed that DIP1 mRNA was unchanged in *dsn* mutant ovaries as compared to controls (Fig 3A). Therefore, we asked if Dsn is required for the localization of DIP1 to satellite bodies. By staining control and *dsn* mutant ovaries with an antibody against DIP1, we did not observe any difference in the localization of DIP1 to the satellite bodies (Fig 3B). This result indicates that Dsn is not required for the localization of DIP1 to satellite bodies.

One possible mechanism is that Dsn could inhibit the binding of DIP1 to nascent transcripts to prevent them from being degraded. Alternatively, Dsn may regulate the activity of DIP1 on nascent transcripts. We performed RNA-immunoprecipitation (RNA-IP) using DIP1 antibody and found that DIP1 generally binds more strongly to INE-1 containing *CG32000*,

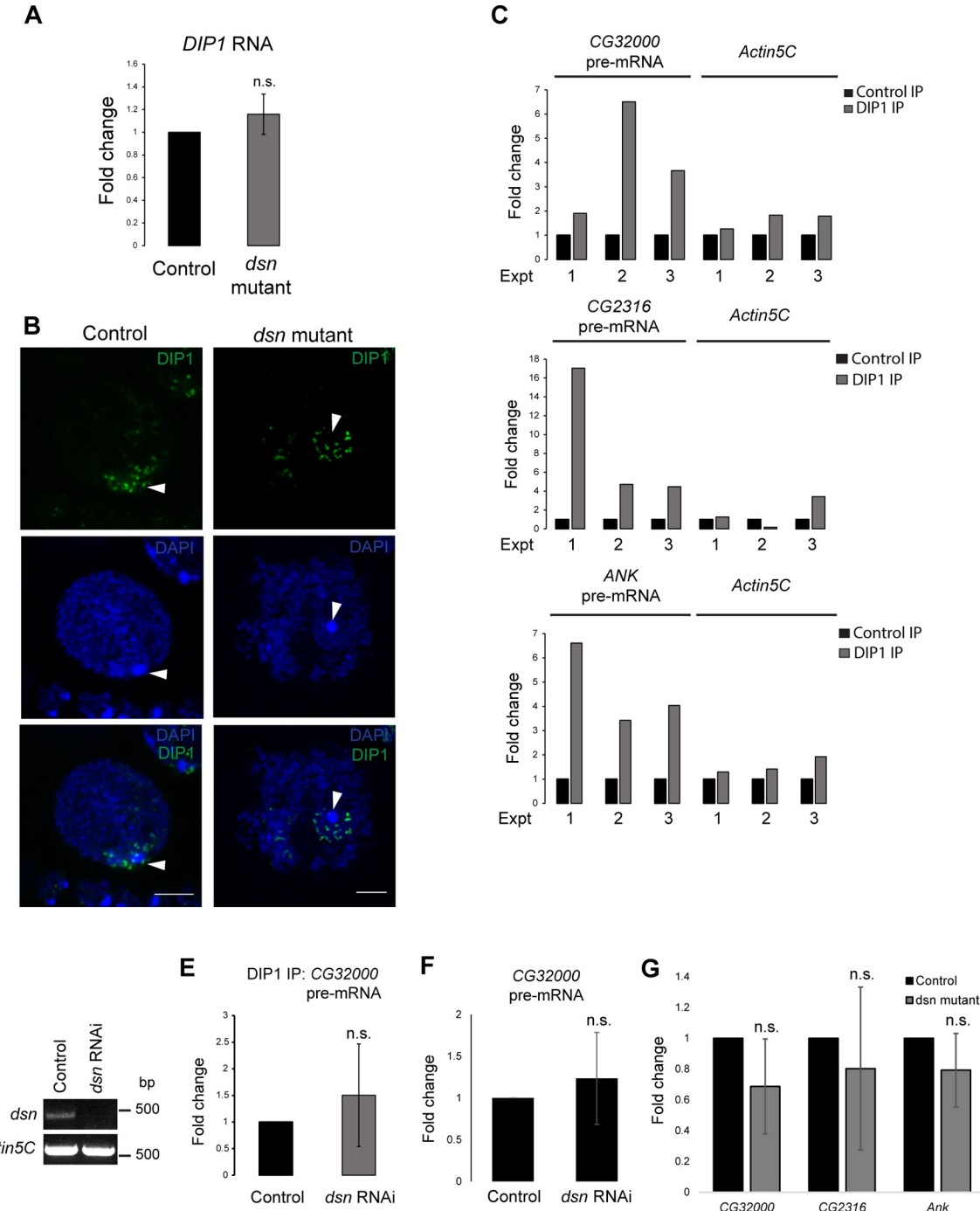

**Fig 3. DIP1 binds to nascent transcripts in a Dsn-independent manner.** (A) RT-qPCR showing no change in the expression of DIP1 mRNA in *dsn* mutant versus control ovaries. N = 3 biological replicates. (B) Confocal images showing the localization of DIP1 (green) in the control and *dsn* mutant nurse cell nuclei. Scale bar: 7 μm. Arrowheads point to the heterochromatin of the fourth chromosomes. (C) RT-qPCR showing enrichment of *CG32000*, *CG2316* and *ANK* pre-mRNAs in three independent DIP1 immunoprecipitation experiments. *Actin5C* was used as a negative control for non-specific pull down. (D) RT-PCR showing the knockdown efficiency of *dsn* in *dsn* RNAi versus control S2 cells. *Actin5C* was used as a loading control. (E) RT-qPCR showing no change in the binding of DIP1 to *CG32000* pre-mRNA in *dsn* RNAi versus control S2 cells. N = 3 biological replicates. (F) RT-qPCR showing no change in the expression of *CG32000* pre-mRNA in *dsn* RNAi versus control S2 cells. N = 3 biological replicates. (G) RT-qPCR showing no change in the binding of DIP1 to *CG32000*, *CG2316 and Ank* pre-mRNA in *dsn* mutant versus control ovaries. N = 3 biological replicates.

*CG2316* and *ANK* pre-mRNAs than to a control *actin5C* mRNA (Fig 3C). As a positive control, we detected an enrichment of INE-1 sisRNAs in the precipitates in S2 cells (S3 Fig). This result was surprising as it showed that nascent transcripts are already bound to DIP1 under normal conditions. We next knocked down the expression of *dsn* by RNAi in S2 cells (Fig 3D) and asked if it led to an increase in the binding of DIP1 to nascent transcripts. To our surprise, we did not observe any changes in the binding of DIP1 to *CG32000* pre-mRNA (Fig 3E). To determine if Dsn regulates nascent transcripts in S2 cells, we examined the expression of *CG32000* pre-mRNA and found no change in its expression in *dsn* RNAi S2 cells (Fig 3F). Therefore, Dsn appears to be dispensable for the regulation of nascent transcripts in S2 cells.

We therefore repeated the DIP1 immunoprecipitation experiments using ovarian lysates where Dsn activity was needed for pre-mRNA stability. Similar to the results from S2 cells, we did not observe an increase in the binding of DIP1 to pre-mRNAs (*CG32000*, *CG2316* and *Ank*) in *dsn* mutant ovaries when compared to wild-type controls (Fig 3G). Together, our results suggested that under normal circumstances, nascent transcripts are already bound by DIP1 independently of Dsn.

## Dsn counteracts the activity of DIP1

We next considered the alternative hypothesis that Dsn may regulate the activity of DIP1 on nascent transcripts. On the FlyBase, DIP1 was found to interact with Lesswright (Lwr) via a yeast-two-hybrid screen. *Drosophila lwr* encodes for the Ubc9 protein, which is a Sumo conjugating enzyme responsible for sumoylation of its targets [20,21]. In *Drosophila*, sumoylation has been found to regulate the activities of various proteins in a dynamic manner [22]. To investigate if DIP1 is sumoylated in vivo, we immunoprecipitated DIP1 in ovarian and S2 cell lysates, and performed western blotting using an antibody detecting Sumo protein. In both lysates, we detected a specific band of ~55 kDa, which is a size that is consistent with sumoylated DIP1 (DIP1: 44 kDa + Sumo: 10 kDa = 55 kDa) (Fig 4A, arrowhead), indicating that DIP1 is indeed sumoylated in vivo. Interestingly, immunostaining of ovaries revealed that Sumo and DIP1 co-localized at the satellite bodies (Fig 4B), suggesting that the activity of DIP1 at the satellite bodies may be regulated by sumoylation.

By performing western blotting, we observed that the majority of the DIP1 protein was not sumoylated in ovaries (Fig 4C, ~44 kDa predicted size), however some DIP1 protein also appeared as slower-migrating (presumably sumoylated) forms (Fig 4C, arrowhead). The appearance of these slower-migrating DIP1 was reduced in *lwr* and *smt3* heterozygous mutant ovaries, indicating that they represent sumoylated forms of DIP1 (Fig 4C). The non-sumoylated form of DIP1 protein was unchanged in *dsn* mutant ovaries (Fig 4D, arrow, ~44 kDa predicted size). In two out of three biological replicates, the abundance of the sumoylated forms of DIP1 was up-regulated in *dsn* mutant ovaries (Fig 4D). We believe that the extent of up-regulation of sumoylation is greater in Expt 2 and 3 than in Expt 1. In Expt 1, the upper-most band in the *dsn* mutant lane was stronger than that in the control lane, suggesting that DIP1 is more heavily modified in the mutants than controls. Thus, the result is still consistent with that of Expt 2 and 3.

In contrast, we observed little change in sumoylated DIP1 in S2 cells after *dsn* RNAi (Fig 4E). This could be due to the fact that majority of the DIP1 was already sumoylated in S2 cells (Fig 4E), explaining why we did not observe any change in *CG32000* pre-mRNA after Dsn knockdown (Fig 3F). Taken together, our data indicated that Dsn represses the sumoylation of DIP1 at the protein level.

To confirm that the decrease in nascent transcripts in *dsn* mutants was indeed caused by the increase in activity of DIP1, we asked if reducing a copy of DIP1 was able to rescue the

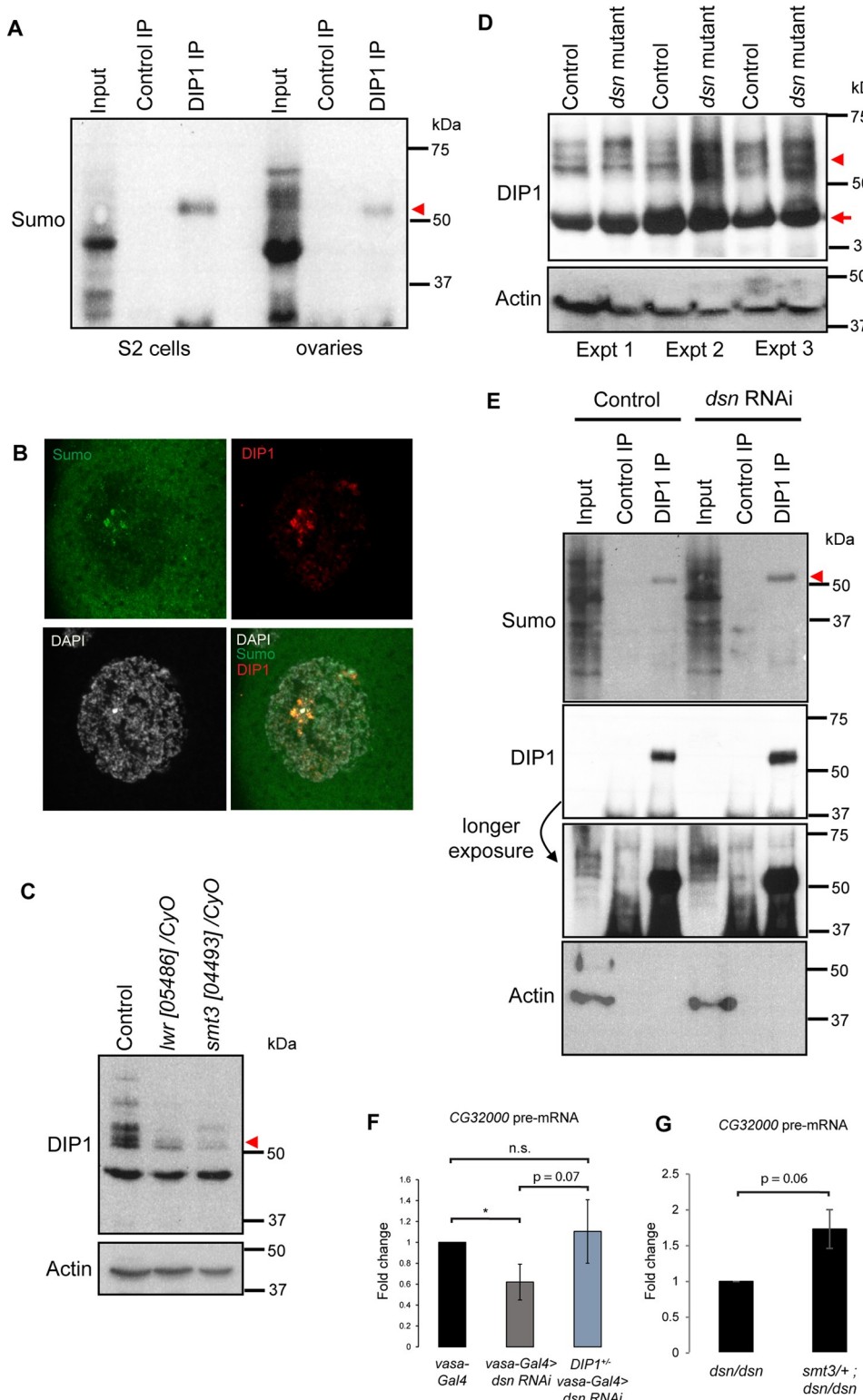

**Fig 4. Dsn counteracts the activity of DIP1 via repressing DIP1 sumoylation.** (A) Western blots showing the presence of sumoylated DIP1 after immunoprecipitation of DIP1 in ovaries and S2 cells. (B) Confocal images showing the enrichment of Sumo proteins (green) at the satellite bodies, co-localizing with DIP1 (red) in wild type ovaries. DAPI (white). (C) Western blot showing the reduction of slower-migrating forms of DIP1 in *lwr/CyO* and *smt3/CyO* ovaries compared to control. Actin5C was used as a loading control. (D) Western blot showing an increase in the

abundance of slower-migrating DIP1 protein in *dsn* mutant versus control ovaries in 3 biological replicates. Actin5C was used as a loading control. (E) Western blots showing the increase in sumoylated DIP1 (arrowhead) in *dsn* RNAi S2 cells. Actin was used as a loading control in the input lanes. (F) RT-qPCR showing downregulation of *CG32000* pre-mRNA in *vasa-Gal4>dsn RNAi* versus *vasa-Gal4* and no change in *CG32000* pre-mRNA expression in *vasa-Gal4* versus *DIP1+/-; vasa-Gal4>dsn RNAi*. *p < 0.05, two-tailed t test. N = 3 biological replicates. (G) RT-qPCR showing upregulation of *CG32000* pre-mRNA in *smt3/+;dsn/dsn* versus *dsn/dsn*. Two-tailed t test. N = 2 biological replicates.

phenotype in *dsn* RNAi ovaries. Consistent with *dsn* mutant ovaries, we also observed a decrease in the expression of *CG32000* pre-mRNA in *vasa-Gal4>dsn RNAi* ovaries (Fig 4F). As expected, reducing a copy of DIP1 was able to rescue the expression of *CG32000* pre-mRNA back to wild-type levels (Fig 4F, no significant difference between *vasa-Gal4* and *DIP1 +/-; vasa-Gal4>dsn RNAi*). To further confirm if the activity of Dsn is due to increase in sumoylation of DIP1, we reduced a copy of *smt3* in the *dsn* homozygous mutant ovaries. Indeed, the level of *CG32000* pre-mRNA was rescued (Fig 4G). Thus, we concluded that Dsn regulates pre-mRNA by counteracting sumoylation of DIP1.

## Discussion

SON encodes for an evolutionary conserved protein that binds to nascent transcripts and localizes to nuclear speckles [23–27]. Various functions have been assigned to SON, which includes regulation of splicing and transcription [23,24,28,29]. Besides that, SON has been implicated in various processes such as stem cell self-renewal and differentiation and cell cycle progression [16,23,25,29]. In human, mutations in SON have been linked to brain developmental defects and leukemia [28,30,31].

In this study, we uncovered a novel role for *Drosophila* homolog of SON (Dsn) in protecting nascent transcripts from unproductive degradation by DIP1 (Fig 5). During transcription, nascent transcripts that contain INE-1 sequences in the introns are bound by both Dsn and DIP1. Dsn inhibits the activity of DIP1, and shields the nascent transcripts from entering the decay pathway. Upon splicing, the intron is excised and Dsn would be released from the transcripts, which leads to the alleviation of the inhibition by Dsn. As a consequence, the INE-1 sisRNA is degraded by DIP1. Conceptually, Dsn acts as a 'timer' to ensure that nascent transcripts are fully spliced before the decay activity of DIP1 kicks in. Based on the fact that SON is highly conserved in mammals and mammalian SON had been shown to bind to nascent transcripts, we envision that this novel role of SON may be conserved.

Our model is consistent with the prevailing idea that nascent transcripts are highly susceptible to decay, and cells have evolved active mechanisms to safeguard nascent transcripts [1]. One such example is the process of telescripting whereby U1 snRNA protects the nascent transcript from cryptic intronic cleavage and polyadenylation [32].

How does Dsn regulate the activity of DIP1? We suggest that Dsn regulates the activity of DIP1 via its sumolyation. In *dsn* mutants, there is a strong correlation between the increase in sumoylated DIP1 and increase in DIP1 activity. This observation suggests that DIP1 activity may be influenced by sumoylation. Our data suggest that sumoylation is the primary modification that is responsible for the subsequent modification of DIP1. We show that reduction of sumoylation pathway genes (*lwr* and *smt3*) led to an overall reduction of DIP1 modification (Fig 4C). Thus, without sumoylation, other forms of posttranslational modifications are also dramatically reduced.

We hypothesize that the satellite body serves as a protective zone where nascent transcripts are bound and protected by Dsn. An interesting hypothesis is that Dsn limits the concentration of modified forms (including sumoylation) of DIP1 so that it is not sufficient to degrade transient nascent transcripts before splicing occurs. After splicing, the introns are released

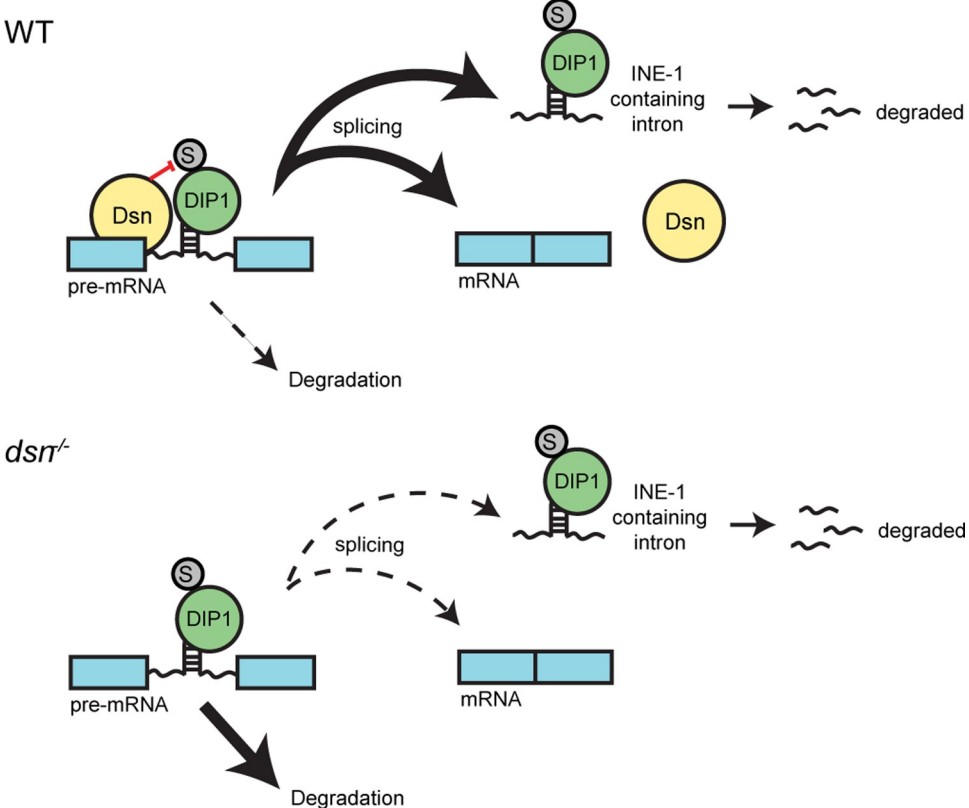

**Fig 5. Working model of Dsn and DIP1 in regulating the abundance of *INE-1* containing pre-mRNA and *INE-1* sisRNA in *Drosophila*.** "S" stands for sumoylation.

from the pre-mRNAs and the repressive effect of Dsn is alleviated. Sumoylation of DIP1 then activates its degradation activity. Sumoylation of DIP1 may influence its conformation and binding partners, thereby modulating its activity [33]. Future work will aim to address the molecular mechanism on how DIP1 activity is regulated by sumoylation.

Although we did not observe any differences in binding of DIP1 to nascent transcripts between wildtype and *dsn* mutant ovaries, we cannot totally exclude the possibility that Dsn may regulate the binding activity of DIP1 to nascent transcripts. As DIP1 binds to and degrades nascent transcripts, we envision the regulation of DIP1 binding and decay as a dynamic and coordinated event. Thus, it is possible that the binding of DIP1 to nascent transcripts may be coupled to its decay activity.

One interesting observation was that the repressive activity of Dsn on DIP1 was seen in ovaries but not in S2 cells. Knockdown of *dsn* in S2 cells did not lead to a dramatic increase in sumoylation of DIP1. This observation can be explained by the fact that most of the DIP1 protein is already being sumoylated in S2 cells, unlike the case in the ovaries. This difference between S2 cells and ovaries may be due to an intrinsic difference in the activity of sumoylation pathway between these two cell types, thus making the ovaries more sensitive to changes in Dsn activity.

In closing, our work encourages the use of satellite body as a model for studying RNA metabolism. We envision that by identifying and characterizing more proteins/RNAs that localize to the satellite body, we can in principle learn more about the intricate regulation of RNA splicing and decay pathways.

## Materials and methods

### Fly strains

*y w* flies were used as controls unless otherwise stated. The following strains were used in this study: *MTD-Gal4* [34], *FLAG- Dsn* (this study), *CG8273$^{GS7314}$* (*dsn* mutant) (Kyoto #201169), *vasa-Gal4* (kind gift from Yukiko Yamashita), *CG8273/dsn* RNAi (TRiP HMS00114 Bloomington #34805), *DIP1[EY02625]*, *smt3$^{04493}$/CyO* (Bloomington #11378) and *lwr$^{05486}$/CyO* (Bloomington #11410). Flies were maintained at 25˚C. Before dissection, newly eclosed females were fed with wet yeast paste for 3 days at 25˚C. Generation of *UASp-FLAG-Dsn* transgenic flies was performed as previously described [15]. PCR of *dsn* full-length coding sequence (CDS) was performed using primers, CACC-dsn Fw (5'CACCATGACGGAGAACACAGA-GAAAGGG3') and dsn Rv (5'CTAGCTGGGCGGAAGAATGCCTAA3'). Transgenic flies were generated by BestGene using P-element-mediated insertion.

### Immunostaining

Immunostaining was performed as described previously [15]. Ovaries were fixed in 16% paraformaldehyde and Grace's medium at a ratio of 2:1 for 10–20 min, rinsed and washed in PBX three times for 10 min each and pre-absorbed in PBX containing 5% normal goat serum for 30 min. Ovaries were incubated in primary antibodies overnight at room temperature, washed in PBX three times for 20 min each and incubated in secondary antibodies for 4 hr at room temperature. Finally, the ovaries were washed again in PBX three times for 20 min each before mounting on slides. The primary antibodies used were rabbit anti-DIP1 (1:300) [15], mouse anti-FLAG (1:500, M2 Sigma) and mouse anti-Sumo (1:500, DSHB 8A2). Images were taken with a Leica SP8 Inverted STED microscope and processed using Leica microscope software, LAS X.

### Actinomycin D treatment

Actinomycin D treatment was performed as described previously [9]. Dissected ovaries were incubated in Grace's medium containing 20 μg/ml actinomycin D with constant rocking at room temperature.

### α-amanitin treatment

Dissected ovaries were incubated in Grace's medium containing 20 μg/ml of α-amanitin with constant rocking at room temperature.

### RNA extraction

Tissues were homogenized in 1.5 ml Eppendorf tubes using a plastic pestle and RNA was extracted using the TRIzol extraction protocol (Ambion) or the Direct-zol RNA miniprep kit (Zymo Research). RNA was quantified using a Nanodrop spectrophotometer to ensure equivalent loading for subsequent experiments.

### RT-PCR

For standard RT-PCR, total RNA was reverse transcribed with random hexamers for 1hr using M-MLV RT (Promega). PCR was carried out using the resulting cDNA. For quantitative PCR (qPCR), SYBR Fast qPCR kit master mix (2X) ABI Prism (Kapa Biosystems, USA) was used and carried out on the Applied Biosystems 7900HT Fast Real-Time PCR system. Primers sequences for INE-1, CG32000 exon, CG2316 exon and DIP1 were reported previously [15].

For calculation of fold-change between controls and mutants/RNAi samples, changes in gene expression were normalized against *actin5C* as a loading control. For calculation of fold-change between DIP1 immunoprecipitations, the abundance of RNA in the immunoprecipitates was normalized against the abundance of the same transcript in the inputs. Primer sequences are CG32000 intron Fw (5' TGGCAACAGTGTCCCAATTA3'), CG32000 exon Rv (5' TGACGCCACCAATGTAACAC3'), CG2316 intron Fw (5' TCTTTTATTTG-GAATGCGTTTCT3'), CG2316 exon Rv (5' ACCCATATCTATTTGCTTCTTCC3'), ANK exon Fw (5' TGATGTGACCCCATTACACG3'), ANK exon Rv (5' TGCATCGCAATTTCCA-GATA3'), ANK exon Fw2 (5' CGATTCCGATGACGAATCTT3'), ANK intron Rv2 (5' GAT-CAATTTCGGACGTCACC3'), Myoglianin intron Fw (5' GGCCCGATTTGGTTATAGGT3'), Myoglianin exon Rv (5' AAAACATCGACCTTGC-GATT3'), Maverick intron Fw (5' TGCCAAACCGTATACAGAAGG3'), Maverick exon Rv (5' AGGAAGCTCCCATGAAGTTG3').

### Western blot

Western blotting was performed as previously described [11,15]. Ovaries were dissected in Grace's medium and homogenized in 2X sample buffer containing β-mercaptoethanol. Primary antibodies used were rabbit anti-DIP1 (1:5000) [15], mouse anti-Sumo (1:1000, DSHB 8A2) and mouse anti-Actin (1:100, DSHB JLA20). Western blot detection was done digitally using the ChemiDoc Touch Imaging System (BioRad) and under non-saturating conditions.

### Immunoprecipitation

S2 cells, which were obtained from Steve Cohen's laboratory, were grown in serum-free medium. ~200 ovaries were dissected for each immunoprecipitation experiment. Immunoprecipitation was performed as previously described [15]. Cells or ovaries were lysed in protein extraction buffer (50mM Tris-HCl pH 7.5, 150mM NaCl, 5mM MgCl$_2$, 0.1% NP-40) supplemented with Protease Inhibitor Cocktail (Roche). Lysates were blocked using protein A/G agarose beads (Merck Millipore) before incubating in rabbit anti-DIP1 (10 μl) overnight at 4°C. As a control, no antibody was added. Protein A/G agarose beads were then added and incubated for another 2 hr. After incubation, beads were washed in protein extraction buffer three times. Protein and RNA were extracted using 2X sample buffer containing β-mercaptoethanol and Direct-zol RNA miniprep kit (Zymo Research), respectively.

### siRNA-mediated knockdown

For dsRNA knockdown, experiments were performed as described previously [35]. S2 cells were treated with 15 μg of dsRNA once for four days before cells were harvested. Primers used for generating the template for in vitro transcription were reported previously [16].

### Supporting information

**S1 Fig. FLAG-Dsn transgene produces a fully functional protein.** (A) RT-qPCR showing over-expression of *dsn* mRNA in *MTD>FLAG-Dsn* versus *MTD/CyO* ovaries. N = 3 technical replicates. (B) *CG32000* pre-mRNA expression was rescued in the *dsn* mutant rescue ovaries as compared to *dsn* mutant. A total of 2 independent experiments were done.
(TIF)

**S2 Fig. Stability of *Marverick* pre-mRNA is not affected in *dsn* mutant ovaries.** Chart showing the levels of *Marverick* pre-mRNA in the ovaries of control and *dsn* mutant flies before and

after 0.5h of α-amanitin treatment. Error bars depict SD from three biological replicates.
(TIF)

**S3 Fig. Positive control for DIP1 immunoprecipitation.** Western blot showing enrichment of DIP1 in DIP1 immunoprecipitate in S2 cells. RT-PCR depicting enrichment of INE-1 sisRNA in DIP1 immunoprecipitate.
(TIF)

## Acknowledgments

We thank Joel Wong Jie Feng for generating FLAG-Dsn transgenic flies, and Kimberly Rae Guzman Peralta for initial characterization of FLAG-Dsn localization. We also thank the Pek laboratory members Chan Seow Neng, Ismail Osman and Rayhan Al-Awdady Binte Ismail for discussion, Yukiko Yamashita, Developmental Studies Hybridoma Band, Kyoto Stock Center and Bloomington Stock Center for providing materials.

## Author Contributions

**Conceptualization:** Jun Wei Pek.

**Investigation:** Mandy Li-Ian Tay, Jun Wei Pek.

**Supervision:** Jun Wei Pek.

**Writing – original draft:** Mandy Li-Ian Tay, Jun Wei Pek.

**Writing – review & editing:** Mandy Li-Ian Tay, Jun Wei Pek.

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
