## [Decision Letter · Decision Letter 0]

15 Oct 2019

Dear Dr Pek,

Thank you very much for submitting your Research Article entitled 'SON protects nascent transcripts from unproductive degradation by counteracting DIP1' to PLOS Genetics. Your manuscript was fully evaluated at the editorial level and by three independent peer reviewers. The reviewers appreciated the attention to an important topic but identified some aspects of the manuscript that should be improved.

We therefore ask you to modify the manuscript according to the review recommendations before we can consider your manuscript for acceptance. Your revisions should address the specific points made by each reviewer.

[LINK]

Yours sincerely,

Gregory P. Copenhaver

Editor-in-Chief

PLOS Genetics

Gregory Barsh

Editor-in-Chief

PLOS Genetics

Reviewer's Responses to Questions

**Comments to the Authors:**

Reviewer #1: My remaining concerns have been satisfactorily addressed in this second revision. I agree with the authors that removal of the data with the Dsn antibody (Figure S2, with which I had questions about validity) does not affect the main conclusions of the paper.

Reviewer #2: Here are what would be my major highlights of this work :

- In drosophila ovaries, Dsn localizes to the satellite body where active decay of INE-1-containing introns by DIP1 occurs after splicing.

- Dsn is a negative regulator of DIP1 activity via inhibition of sumoylation and, likely, of subsequent post-translational DIP1 modifications.

- This Dsn-mediated DIP1 inhibition protects the spliced introns less efficiently than the corresponding pre-mRNAs.

I think these data warrant a paper in PLOS Genetics but I would like to warn the reader against some overstatements found in the Abstract and working model (Figure 5):

- I am not convinced that “nascent transcripts are already bound by DIP1 independently of Dsn” (page 9 bottom of first paragraph)

- There is no evidence either that, page 11 :” Upon splicing, (…) Dsn would be released from the transcripts, “

A possible alternative version of the Abstract would be :

Gene expression involves the transcription and splicing of nascent transcripts through the removal of introns. In Drosophila, a double-stranded RNA binding protein Discointeracting protein 1 (DIP1) targets INE-1 stable intronic sequence RNAs (sisRNAs) for degradation after splicing. How nascent transcripts that also contain INE-1 sequences escape degradation remains unknown. Here we observe that these nascent transcripts can also be bound by DIP1 but the Drosophila homolog of SON (Dsn) protects them from unproductive degradation in ovaries. Dsn localizes to the satellite body where active decay of INE-1 sisRNAs by DIP1 occurs. Moreover, the pre-mRNA destabilization caused by Dsn depletion is rescued in a DIP1 heterozygous mutant, suggesting that Dsn is a negative regulator of DIP1. In fact, Dsn is a repressor of DIP1 posttranslational modifications (primarily sumoylation) that are assumed to be required for efficient DIP1 activity. This Dsn-mediated reduction of DIP1 posttranslational modifications is not sufficient to totally abolish DIP1-mediated sisRNAs decay.

Minor comments:

- Figure 2F: Dsn should also point positively toward INE-1 introns (using a broken arrow ?) to emphasize the fact that the amount of INE-1 is also somewhat reduced in the absence of Dsn (see Figure 2B)

- Since Figure 1D was removed, Figure 3D should be removed too.

- I could not find how Dsn RNAi was performed in ovaries (Figure 3)

Reviewer #3: This manuscript describes a process by which pre-mRNAs carrying introns that are potential targets for degradation are protected until splicing has occurred. The authors primarily focus on the Drosophila ovary, where they have previously implicated the proteins DIP1 and Dsn (the Drosophila homolog of SON) as regulators of gene expression. They report that a FLAG-tagged version of Dsn localizes to the satellite body (a nuclear body where DIP1 also localizes), and a loss of function mutation in Dsn results in loss of RNA that carries INE-1 elements, which is opposite to the phenotype of loss of function mutations in DIP1, and suggests a protective role for wild type Dsn. They hypothesize that the two proteins act in opposition via Dsn regulation of the RNA-binding DIP1. RNA-IP experiments with a DIP1 antibody support that DIP1 binds to INE-1 RNAs more strongly than to other RNAs, and, in Dsn mutants, DIP1 binding to INE-1 pre-RNAs is unchanged relative to wild type, suggesting that Dsn may regulate DIP1 activity rather than RNA-binding. The authors use western blot analysis to support that DIP1 is sumoylated, likely by Lwr since modified DIP1 is decreased in oocytes heterozygous for an allele of lwr, and that SUMOylation of DIP1 increases in a Dsn mutant background. This leads to a model wherein DIP1 bound to pre-mRNAs carrying INE-1 sequences is held inactive by a SUMOylation-inhibiting activity of Dsn; after splicing, Dsn would no longer inhibit SUMOylation, which would lead to activitation of DIP1 activity and subsequent degradation of INE-1 introns.

The overall story is a very interesting example of RNA biology - how nascent transcripts can be protected when spliced introns are targeted for degradation is likely to be of broad interest to the readership of PLoS Genetics, and the paper addresses some specific mechanistic insights. I am overall positive on the paper and commend the authors on pursuing this interesting biology, and I recognize that the authors have already been through peer review at another journal. However, I believe that several aspects of the manuscript should be addressed before publication. My specific comments follow.

Comments that may require additional evidence/analysis:

1. for the super-resolution deconvolution microscopy, it would be helpful to state the specific type of microscopy in the main text (I believe it is STED). In addition, the analysis of overlap performed in the figure is not described in the main text or methods - are these single optical slices? Were they chosen for any specific reason, or are the “representative”? In addition, some summary of the percent overlap of the two signals overall would be helpful in addition to the examples shown. Finally, I am unclear on the authors’ conclusions from this experiment - they refer to DIP1 and FLAG-Dsn as forming an “intermingling network” - what does this phrase mean?

2. in Figure 3C, please use statistical tests to compare binding of DIP1 to INE-1 RNAs relative to the Act5C control RNA. (It is surprising that statistics are used in most other comparisions in the paper, but not here.)

3. the qRT-PCR experiments in figure 2b and c are well controlled with analyses of genes that do not harbor INE-1 elements. The experiments in figure 2d and e are not controlled - the same analysis (at least one) should be done with a gene that does not harbor an INE-1 element (e.g. Myoglianin or Maverick) to show that it the effect is specific to the proposed mechanism.

4. The authors state that their western exposures are non-saturating, but the 44kDa DIP1 bands in Figure 4d do appear saturated, making comparisons between lanes challenging. Since Figure 4f represents the only experiment tying dsn to their proposed model of DIP1 regulation by SUMOylation, ideally these data would be better presented and more convincing, particularly since at least one trial shows no clear evidence of a change. Similarly, ideally these data would be quantified and compared statistically.

5. the final experiment in figure 4f confirms that a loss of a copy of DIP1 suppresses the dsn mutant phenotype on CG32000 pre-mRNA levels (importantly, it also provides a second dsn mutant genetic background to confirm that the phenotype is not specific to the one allele used thus far - the authors may wish to highlight this, as it strengthens their evidence). While this is an important and interesting result, it does not speak to the proposed model of DIP1 regulation by SUMOylation - i.e. there is no evidence that a change in DIP1 SUMOylation regulates its activity. This, to me, is a key experiment to tie the SUMOylation data to the rest of the manuscript. Could the authors, for example, examine INE-1 pre-RNAs in the lwr background where DIP1 sumoylation is significantly reduced? It seems that all of the necessary reagents and techniques are in place for this experiment, and there is a clear prediction. Genetic evidence for regulation via DIP1 SUMOylation would solidify their proposed model; in the absence of this evidence, the data are largely circumstantial.

Comments that can be addressed via changes to the text:

1. the authors state that in their dsn mutants, there is a decrease in INE-1 sisRNAs and mRNAs from genes harboring INE-1 (figures 2b-c). This is based on assessment of three genes, CG32000, CG2316, and ANK. Exonic qRT-PCR shows that ANK levels are not reduced, even though pre-mRNAs for this gene are reduced. The authors should be clear that 2 of 3 genes follow the pattern they describe, and comment on why this might be the case.

2. for Figure 4a, please describe what the “Control IP” is (e.g. in the methods or legend).

3. the localization of the satellite body to the region of the 4th chromosome is based on DAPI staining only, where a high density of localized staining is presumed to represent the 4th. Since the chromosome is not specifically identified (e.g. through DNA-FISH or POF staining), it seems that wording like “the presumed 4th” would be more appropriate.

4. I am confused by the presentation of data from S2 cells. The authors first report “an enrichment of INE-1 sisRNAs in the precipitates in S2 cells”, which is presented as a positive control of RNA-IP experiment in oocytes. It is unclear to me how S2 cells are a positive control in this experiment (I presume “precipitates” refers to RNA-IP with DIP1 antibody.) They then report the a Dsn knockdown in S2 cells does not change DIP1 binding to CG32000, nor is there a change in CG32000 pre-mRNA in S2 cells - whereas there was a change in oocytes. Finally, they show no change in DIP1 sumoylation in dsn knockdown cells. It is unclear how the negative data from S2 cells help to explain the relationship between Dsn and DIP1, except that perhaps the relationship is specific to the ovary. Is that the point? If so, perhaps the relevance of this could be commented on in the Discussion.

5. While the experiments and phenomenology are quite interesting, I confess that I found this paper very challenging to read and understand. I see from prior reviews that a previous reviewer made similar comments and suggestions for improving the manuscript, which the authors opted not to do. Here are specifics that I hope the authors can improve upon:

-it is confusing that the authors are inconsistent in whether they use SON or Dsn - the title uses SON, but most of the Results refer to Dsn, but then the authors revert to SON again in the Discussion. It would be helpful to stick with one name after telling us about both names.

-the introduction is very short, and I did not understand the background information until I had read the whole manuscript. It would be helpful to better explain the background information in the Introduction. Specifically, what it INE-1? and, explain sisRNAs - they are “stable”, but they are degraded? Does the term refer to the pre-mRNAs, the spliced introns, or both, or something else? I don’t understand the name. And, how are satellite bodies “microscopically visible”? by staining? (for what?) or by some other means? Also, it becomes evident later that satellite bodies localize to the 4th chromosome region, it would be helpful to know this in the introduction.

-I do not understand the statement “nascent transcripts are already susceptible to degradation by binding to DIP1” in the Introduction. I wonder whether this simply means “nascent transcripts are bound to DIP1”, or is there more to this?

-the results begin with reference to the gene rga - what is this gene, and how is it relevant to the present study? (is it a sisRNA? a satellite body component? Is it perhaps not relevant?)

-in describing the localization of FLAG-Dsn, I appreciate the honesty of the authors, but I believe they do themselves a disservice by explaining that the localization of an overexpressed protein may not reflect the endogenous localization before the actual localization is even discussed. I understand that this was added in response to a previous reviewer, but as written, it undermines the confidence of the reader - if the authors do not believe that the FLAG-Dsn localization is valid, the data should not be in the paper. If they do think it is valid, the language of the paragraph should reflect why they are confident that it is so, even with a potential caveat.

-in the Discussion, the authors state that nascent transcripts are bound by Dsn. I do not see the evidence for this in the current manuscript, a citation for this result should therefore be provided (or please correct my error if the data are in fact included).

**Have all data underlying the figures and results presented in the manuscript been provided?**

Reviewer #1: Yes

Reviewer #2: Yes

Reviewer #3: Yes

PLOS authors have the option to publish the peer review history of their article (what does this mean?). If published, this will include your full peer review and any attached files.

Reviewer #1: No

Reviewer #2: No

Reviewer #3: No

---

## [Decision Letter · Decision Letter 1]

28 Oct 2019

Dear Dr Pek,

We are pleased to inform you that your manuscript entitled "SON protects nascent transcripts from unproductive degradation by counteracting DIP1" has been editorially accepted for publication in PLOS Genetics. Congratulations!

Yours sincerely,

Gregory P. Copenhaver

Editor-in-Chief

PLOS Genetics

Gregory Barsh

Editor-in-Chief

PLOS Genetics

Comments from the reviewers (if applicable):

Reviewer's Responses to Questions

**Comments to the Authors:**

Reviewer #3: The authors have addressed my primary concerns.

Note that, for one experiment that they added in response to one of my comments (Fig 4g), a difference was noted with a p value of 0.06. Some may take issue with this result as "not significant". However, I don't believe that a p value of 0.05 has any magical properties, though it is often used as a standard to determine whether a biological effect is "real". It is likely that a third replicate of this experiment would provide more statistical power to reach the common 0.05 threshold, and in a perfect world these authors would perform this experiment. But, I am fine with the current practice of the authors - i.e. simply report the magnitude of the difference and the associated p value for the two replicates that they performed.

**Have all data underlying the figures and results presented in the manuscript been provided?**

Reviewer #3: Yes

PLOS authors have the option to publish the peer review history of their article (what does this mean?). If published, this will include your full peer review and any attached files.

Reviewer #3: No

**Data Deposition**

http://datadryad.org/submit?journalID=pgenetics&manu=PGENETICS-D-19-01345R1

**Press Queries**

---

## [Editor Report · Acceptance letter]

6 Nov 2019

PGENETICS-D-19-01345R1 

SON protects nascent transcripts from unproductive degradation by counteracting DIP1 

Dear Dr Pek, 

We are pleased to inform you that your manuscript entitled "SON protects nascent transcripts from unproductive degradation by counteracting DIP1" has been formally accepted for publication in PLOS Genetics! Your manuscript is now with our production department and you will be notified of the publication date in due course.

With kind regards,

Matt Lyles

PLOS Genetics

On behalf of:
